# Effects of Salt Stimulation on Lunasin Accumulation and Activity during Soybean Germination

**DOI:** 10.3390/foods9020118

**Published:** 2020-01-22

**Authors:** Weiyi Zhang, Yuqiong Hao, Cong Teng, Xin Fan, Xiushi Yang, Mengjie Liu, Guixing Ren, Congping Tan

**Affiliations:** 1College of Food Science and Engineering, Qilu University of Technology (Shandong Academy of Sciences), Jinan, Shandong 250353, China; zhangweiyi312@163.com; 2Institute of Crop Sciences, Chinese Academy of Agricultural Sciences, No. 80 South Xueyuan Road, Haidian, Beijing 100081, China; haoyuqiong334@163.com (Y.H.); 82101175189@caas.cn (C.T.); yangxiushi@caas.cn (X.Y.); liumengjie1009@163.com (M.L.);

**Keywords:** lunasin, soybean, germination, antioxidant activity, anti-inflammatory activity

## Abstract

Lunasin, a bioactive peptide, was originally found in soybeans, and it has exhibited multiple biological functions. On the basis of previous studies, salt stress was found able to induce changes in many polypeptides and translatable mRNA levels in plants. Salt stress was applied to soybean germination, with water treatment as a control group, to evaluate the effects of salt stimulation on lunasin accumulation and activity during soybean germination. Lunasin content gradually increased in the control group during germination, reached the highest level after six hours of imbibition, and then slowly decreased. Under salt exposure, lunasin content showed a similar trend to that of the control group. The lunasin content in salt-treated soybean was significantly higher than that in the control group. Lunasin peptide was purified from soybean after six hours of imbibition and it was then used for function evaluation. Purified lunasin from salt-stress-germinated soybean (6 h-LSGS) exhibited stronger antioxidant activity than lunasin from water-treatment-germinated soybean (6 h-LWGS) and soybean seed without imbibition (DRY). The 6 h-LSGS presented anti-inflammatory activity on LPS-induced macrophage cells (*p* < 0.05) by suppressing the release of nitric oxide (NO) and proinflammatory cytokines, including IL-1 and IL-6. The gene expression of *NOS*, IL-1, IL-6, and TNF-α was significantly inhibited by 6 h-LSGS. Further, 6 h-LSGS exhibited superior antiproliferation activity on human breast-cancer cells MDA-MB-231 when compared to 6 h-LWGS and DRY. Overall, this study offers a feasible elicitation strategy for enhancing lunasin accumulation and its properties in soybean for possible use in functional food.

## 1. Introduction

Lunasin, which is a soybean-derived bioactive peptide, has shown positive effects on many biological functions. Lunasin, originally discovered in soybeans, has a molecular weight of 5.5 KD million and 44 amino acids. Arg, Gly, and Asp residues and the cell-adhesion module that is composed of nine aspartic acid residues at the carboxyl terminal determine the biological activity of lunasin. Lunasin reaches the target organ or tissue through decomposition by the gastrointestinal digestive enzyme, serum protease, and peptidase in the body. Subsequently, it binds to cells through Arg–Gly–Asp, and regulates cell migration, growth, differentiation, and apoptosis [1]. Extensive scientific research has shown that lunasin has natural antioxidant, antiallergic, and anticancer effects, and it helps to regulate cholesterol biosynthesis in vivo [2]. 

Germination is one of the ways to promote the significant accumulation of lunasin in soybeans [3]. In the process of soybean germination, temperature, germination time, and light treatment have been affected to accumulate lunasin [4,5]. Salt stress is considered to effectively promote secondary metabolic biosynthesis in plants, such as phenolic compounds, saponins, alkaloids, and gluconate [6,7]. Previous research showed that salt stress could induce changes in many polypeptides and translatable mRNA levels in plants. Studies on NaCl treatment length, NaCl concentration, salt-stress recovery, and the effects of other stresses showed that these peptides play a special role in plant salt stress [8]. 

Seed germination refers to a series of orderly physiological and morphogenetic seed processes, starting from imbibition. Sometimes, NaCl stress does not induce polypeptides to disappear, or cause the synthesis of unique polypeptides, but it could decrease or increase the synthesis of a number of polypeptides [9]. Félicie et al. compared the patterns of total protein that were extracted from leaves of control and salt-treated plants, and found that a 22 kDa, pI 7.5 polypeptide accumulated when plants were exposed to NaCl [10]. Asian countries have the traditional habit of eating soybean sprouts, and its composition change is of great significance in food processing and consumption. This study not only evaluates the effects of salt stimulation on lunasin content and activity in the soybean germination process, but it also identifies the key germination time point of greater lunasin accumulation by salt stimulation. It was the first time that salt stress was applied to soybean germination, which aimed at increasing lunasin content and activity.

## 2. Materials and Methods

### 2.1. Materials and Reagents

The typical soybean varieties were harvested in northeastern China in 2018, Institute of Crop Sciences, provided by the Chinese Academy of Agricultural Sciences. The lunasin standard was from the Beijing Genomics Institute (Beijing, China). RAW264.7 macrophages and the MDA-MB-231 breast-cancer cell line originated from the Institute of Bioscience, Chinese Academy of Sciences (Shanghai). Penicillin/streptomycin (Invitrogen, Carlsbad, CA, USA), 2,2-azino-bis (3-ethylbenzothiazoline-6-sulfonicacid) diammonium salt (ABTS), 1,1-diphenyl-2-picrylhydrazyl radical (DPPH), fluorescein sodium, and lipopolysaccharide (LPS) were purchased from Baierdi Biotechnology Co., Ltd. Dulbecco’s modified Eagle’s medium (DMEM) was from Thermo Fisher Scientific (Beijing, China) and fetal bovine serum (FBS) was purchased from Sigma-Aldrich (St. Louis, MO, USA). Mass spectrometer (SCIEX TripleTOF6600^®^) Nitrogen generator (SCIEX, Massachusetts, USA). ACQUITY UPLC ^®^BEH Shield RP18 (Waters Corp, Milford, MA, USA).

### 2.2. Soybean Germination 

The soybeans were subjected to two different treatments: germination under water treatment or under salt stress. The seeds were disinfected by immersing in 1% sodium hypochlorite for five min. and rinsing repeatedly with distilled water. Washed high-quality soybean seeds that were the same size and without decolorization were soaked in 25 °C water or 50 mM NaCl solution for 1 h, and then transferred into Petri dishes placed in a thermostat dark house. Germination was carried out in dark conditions at 25 ± 1 °C. Soybean sprouts in each group were collected after 6, 12, 24, 36, and 48 h, and immediately stored at −80 °C.

### 2.3. Protein Extraction and Purification

Phosphate-buffered saline (PBS) was used as extraction solution; the ratio of material to liquid was 1:10 and extraction was at 4 °C for 48 h. The supernatant was obtained by centrifugation. Peptides with a molecular weight of less than 10 KD were obtained by membrane ultrafiltration [11]. We referred to Ren’s method with minor modification for further low-molecular-weight peptide purification [2].

### 2.4. UPLC-MS/MS Analysis

Lunasin was determined in soybeans while using UPLC-MS/MS (ultra performance liquid chromatography/trandem mass spectrometry). Mass spectrometry conditions included electrospray ionization (ESI), negative-ion-detection method, cone hole voltage of 15 V, capillary voltage of 0.8 kV; ion source was 12 °C. The chromatographic column we used was Acquity UPLC ^®^BEH Shield RP18 (2.1 mm × 100 mm, 1.7 μm). Mobile phase: 0.1% formic acid in water (A), 0.1% formic acid in acetonitrile (B), flow rate: 0.2 mL·min.^−1^, column temperature: 30 °C, injector temperature: 10 °C, injection volume: 10 μL. Gradient elution sequence: 1 min., 95% A; 2 min., 65% A; 4 min., 20% A; and, 5 min., 95% A. MS data were collected from 0 to 5 min.

### 2.5. Western Blot and ELISA Detection

Western blot analysis was performed on the basis of a previously published method [12], with minor modifications. The protein sample that was separated by SDS-PAGE (sodium dodecyl sulfate- polyacrylamide gel electropheresis) was transferred to the solid-phase carrier (nitrocellulose film). After rinsing the membrane with PBS for 10 min., the membrane was moved to the 5% skimmed-milk-powder sealing solution that was configured with PBST, and shaken and sealed in the shaker at room temperature for 2 h. The first antibody was diluted with tris-buffered saline containing 0.05% Tween-20 (TBST) to an appropriate concentration, and the membrane was removed from the blocking solution and then placed in the antibody diluent. The shaker was incubated overnight at 4 °C. It was then washed with TBST in a shaker at room temperature three times for 10 min. each time. The diluent of the second antibody was prepared with the same method and it came into contact with the membrane and incubated at room temperature for 2 h. The chemiluminescence reaction was carried out by washing with TBST in a shaker at room temperature three times for 10 min. each time [13].

The ELISA refers to the Dia method, with minor modifications [14]. The samples were diluted to working concentration with coated buffer, cultured at 37 °C for 3 h, removed the coating solution, and washed four times. The pat protein solutions of different concentrations (0, 1.56, 3.13, 6.25, 12.5, 25, 50, and 100 μg/L) were added to the enzyme plate, each concentration was repeated three times, incubated at 37 ℃ for 0.5 h, and the plate was washed four times. Subsequently, 100 μL McAb solution was added, diluted to the working concentration, cultured at 37 °C for 0.5 h, and the plate was washed four times. After washing the plate, enzyme-labeled sheep antirabbit antibody, 1000-fold diluted, was added and incubated at 37 °C for 0.5 h four times. Finally, 100 μL substrate buffer was added to each well; after 10 min., 50 μL terminating solution was added to each well to measure the OD value of each well at 450 nm wavelength.

### 2.6. Antioxidant-Activity Determination

The scavenging rates of DPPH and ABTS^+^ free radicals were determined with reference to the method in [15,16,17,18], with minor modifications. For DPPH analysis, DPPH was dissolved in methanol, and samples were prepared in a solution with a concentration of 0.125/0.25/0.5/1 mg·mL^−1^. We then placed 2 mL DPPH in the test tube, added 2 mL sample solution, mixed well, avoided light for 1 h, and then determined sample absorbance at 517 nm. For ABTS^+^ analysis, samples were dissolved in ultrapure water, and the final concentration gradient was 0.125/0.25/0.5/1 mg·mL^−1^. We put the ABTS^+^ solution in the test tube, added the sample solution, mixed well, avoided light for 2 h, and then determined the absorbance of the sample at 734 nm [18]. The antioxidant properties of ABTS^+^ radicals are different from those of DPPH. ABTS analysis is superior to DPPH analysis when the sample contains hydrophilic antioxidants [19].

### 2.7. Anti-Inflammatory-Activity Assay

The experiment was carried out according to the Dia scheme [14]. After anti-inflammatory-activity assay, the RAW264.7 cell was collected and used for RNA isolation on the basis of the protocol of the Cell RNA Extraction Kit (TianGen Biotech, Beijing, China); cDNA was synthesized while using a cDNA Synthesis SuperMix Kit (TianGen Biotech, Beijing, China). NOS, IL-1, IL-6, and TNF-α expressions in the RAW264.7 cell were measured through qRT-PCR, which was performed on an ABI 7500 Real-Time System (Applied Biosystems, San Francisco, CA, USA). The mouse actin gene was used as the control to calculate gene expression in the qPCR according to the 2^–ΔΔCt^ method. Table 1 shows all of the primers. 

### 2.8. Anticancer-Activity Assay

MDA-MB-231 cells were incubated in a DMEM medium that was supplemented with 1% penicillin/streptomycin and 10% fetal bovine serum, and filled with 5% CO_2_ cells at 37 ℃. The cells were electroplated in 96 well plates at 2 × 10^4^ cell/hole density for overnight incubation, and treated with lunasin at different concentrations and incubated for 72 h. Afterwards, Hank’s Balanced Salt Solution (HBBS) was added to the cells and placed at 37 ℃ for 1 h. Subsequently, absorbance at 570 nm was calculated by spectrophotometer. 

### 2.9. Statistical Analysis 

All of the experiments were repeated more than three times. The values are expressed as the means of three independent experimenters’ SD (STDEV). GraphPad 5.0 (GraphPad Software Inc., San Diego, CA, USA) and SPSS 17.0 (SPSS Inc., Chicago, IL, USA) were used for statistical analysis. The difference was statistically significant (* *p* < 0.05, ** *p* < 0.01). SPSS analyzed all of the graphical representations.

## 3. Results

### 3.1. Lunasin-Content Detection

Figure 1 shows the expression patterns of lunasin at different soybean germination stages. During the germination of soybean seeds, lunasin bands significantly deepened in hours 0–6, peaking at 6 h, and obviously decreasing thereafter (Figure 1A). Under salt exposure (Figure 1A), the lunasin bands showed similar patterns and were significantly increased in depth compared to the control. This suggests that lunasin content was significantly accumulated after salt treatment, which indicated that it was viable for increasing the content of lunasin in soybean by the salt treatment of the germinating soybeans. 

Lunasin content was measured through ELISA (Figure 1B). The contents of the lunasin peptide in the soybeans were 0.53 mg·g^−1^ (DRY), 0.93 mg·g^−1^ (6 h-LWGS), 0.63 mg·g^−1^ (12 h-LWGS), 0.33 mg·g^−1^ (24 h-LWGS), 0.29 mg·g^−1^ (48 h-LWGS), 2.24 mg·g^−1^ (6 h-LSGS), 0.68 mg·g^−1^ (12 h-LSGS), 0.41 mg·g^−1^ (24 h-LSGS), 0.32 mg·g^−1^ (36 h-LSGS), and 0.22 mg·g^−1^ (48 h-LSGS). 6 h-LSGS led to much higher lunasin content (2.4-fold) when compared to 6 h-LWGS. The change of lunasin content during soybean germination under salt stimulation was recorded for the first time. The polypeptide content increase or decrease under salt stress could be due to altered mRNA processing, transcription regulation, transport, stability, or due to the changed rates of protein degradation [8]. It might also be due to the inhibition or stimulation of mRNA translation to varying degrees by increased cytoplasmic ion (Na and Cl) concentrations [20]. Park et al. found that the lunasin content accumulated during soybean germination, similar to a previous study [3]. Paucar-Menacho et al. also showed that lunasin content increased by 61.7% during soybean germination at 25 ℃ for 42 h [21].

### 3.2. Mass Spectrometry Analysis

UPLC-MS/MS was used to further confirm that lunasin was indeed present in the sample and the ELISA results. The lunasin chromatograms clearly showed a peak at the retention time of 3.66 min. (Figure 2A). The mass spectrum acquired from the peak at 3.66 min. generated [M + 7H]^7+^ at 718.90 m/z, [M + 6 h]^6+^ at 838.54 m/z, [M + 5H]^5+^ at 1006.45, and [M + 4H]^4+^ at 1257.39 m/z (Figure 2B), which was consistent with a previous report [22]. 

### 3.3. Antioxidant Activity Assay

The antioxidant functions of DRY, 6 H-LWGS, and 6 H-LSGS were evaluated by measuring the scavenging activities of DPPH and ABTS^+^ free radicals. The results showed that the antioxidant function of DRY, 6 h-LWGS, and 6 h-LSGS was dose-dependent. In the DPPH radical assay (Figure 3A), the scavenging activity of 6 h-LSGS (IC50, 0.28 mg·mL^−1^) was significantly higher than that of 6 h-LWGS (IC50, 0.57 mg·mL^−1^) and DRY (IC50, 0.76 mg·mL^−1^) at a concentration of 1 mg·mL^−1^. In the ABTS^+^ radical assay (Figure 3B), the IC50 of 6 h-LSGS was 0.12 mg·mL^−1^, which was stronger than that of 6 h-LWGS (IC50 = 0.37 mg·mL^−1^) and DRY (IC50 = 0.48 mg·mL^−1^). Overall, salt treatment improved the antioxidant effect of soybean lunasin extract.

Lunasin can protect Caco-2 cells from oxidative damage caused by hydrogen peroxide and tert-butyl hydroperoxide, similar to the results of our research [23]. These findings confirm that lunasin has effective antioxidant activity. In addition, it was previously shown that lunasin can inhibit experiment cataract induced by d-galactose in rats and upregulate antioxidant enzymes [24]. Ren et al. in vitro studied lunasin antioxidant activity in quinoa [2]. The superior antioxidant effect of the lunasin extract from salt-treated soybean in comparison to that from the control could be ascribed to the higher accumulation of lunasin. However, as salt stress can induce the accumulation of a variety of antioxidants, such as saponins, isoflavones, tocopherols, and carotenoids [25], the synergistic effect of these antioxidant components might contribute to the significant increase in antioxidant activity of salt-stress samples [26].

### 3.4. Anti-Inflammatory-Activity Assay

In this study, the anti-inflammatory function of lunasin was evaluated through monitoring the immune response in LPS-stimulated mouse macrophage 264.7 cells. In the cytotoxicity assay (Figure 4A), after 24 h incubation with 6 h-LSGS, 6 h-LWGS, and DRY, the survival rate of cells did not change much at the tested concentrations (0.25–2 mg·mL^−1^). In the NO assay (Figure 4B), 6 h-LSGS, 6 h-LWGS, and DRY significantly inhibited NO accumulation in the culture medium in a dose-dependent way. A significant increase was noted in the inhibition rate of 6 h-LSGS (70.2%) at 2 mg·mL^-1^ in comparison with 6 h-LWGS (58.47%) and DRY (51.97%). Further, the gene expressions of *NOS*, IL-1, IL-6, and TNF-α were significantly upregulated after LPS induction, as shown in Figure 4C. However, expression levels were highly inhibited by lunasin extract from the soybean. In addition, 6 h-LSGS exhibited a significantly higher inhibition rate than that from 6 h-LWGS. The better anti-inflammatory effect of salt stress in comparison to that of the control could be the result of increasing lunasin content in soybean sprouts after salt treatment. The results suggested that soybean lunasin could inhibit NO accumulation, as well as gene expressions either directly or indirectly associated with inflammation. 

Similar to our study, lunasin was reported to inhibit NOS expression in LPS-induced RAW264.7 macrophage cells, which suggested that lunasin performed its anti-inflammatory activity by regulating the iNOS/NO signal pathway [14]. Blanca et al. investigated the anti-inflammatory activity of lunasin on the mouse macrophage 264.7 cell line, and macrophage cells were not inhibited by lunasin-related fragments, and found that the complete primary sequence of lunasin was needed to reduce the reactive oxygen species (ROS) induced by LPS-induced macrophages [27]. Further, Cam and De Mejia found that lunasin has the potential to inhibit αVβ3 integrin-mediated proinflammatory markers by downregulating the activation of the Akt-mediated NF-κB pathways [28]. 

### 3.5. Anti-MDA-MB-231 Activity Analysis

The effects of lunasin on chemical carcinogens were confirmed in cells in vitro [29,30]. Moreover, lunasin can inhibit the transformation of mammalian cells induced by oncogene E1A, and reduce cancer incidence in mouse models [30]. This study measured the cytotoxic and antiproliferation effects of lunasin purified from soybeans on human breast-cancer MDA-MB-231 cells. The cell cytotoxic assay results showed that lunasin from DRY, 6 h-LWGS, and 6 h-LSGS presented no cytotoxicity to MDA MB-231 cells at concentrations from 0.5 to 2 mg·mL^−1^ (Figure 5A). In the antiproliferation assay (Figure 5B), MDA-MB-231 cell proliferation was highly inhibited by DRY, 6 h-LWGS, and 6 h-LSGS. In addition, 6 h-LSGS (69%) exhibited a significantly higher inhibition rate on cell proliferation than did 6 h-LWGS (52%) and DRY (37%) at a concentration of 2 mg·mL^−1^.

Consistent with our research, Jiang et al. found that lunasin can inhibit the proliferation and differentiation of breast-cancer cells [1]. Lunasin suppressed the metastasis of breast-cancer cells through the inhibition of the NF-κB and FAK/Akt/ERK signaling pathways. Furthermore, Hsieh et al. demonstrated that human estrogen-independent breast-cancer MDA-MB-231 cells are significantly inhibited by lunasin when combined with aspirin when compared with inhibitions after using each compound alone [31].

## 4. Conclusions

This paper evaluated the effects of salt stimulation on lunasin accumulation and activity during soybean germination; 50 mM NaCl was applied to soybean germination, and water treatment was recognized as the control group. The lunasin content gradually increased in the control group during germination, reached the highest level after 6 h imbibition, and then slowly decreased. Lunasin content also exhibited a trend of increasing and then decreasing under salt exposure. The lunasin content in germinating soybeans under salt stimulation was significantly higher than that in the control, and salt stimulation (6 h-LSGS) was 2.5 times that of the control group (6 h-LWGS). Moreover, 6 h-LSGS exhibited stronger antioxidant, anti-inflammatory, and anticancer activity than 6 h-LWGS. Overall, this study offers a feasible elicitation strategy for enhancing lunasin accumulation and its properties in soybeans for possible use in functional food.

## Figures and Tables

**Figure 1 foods-09-00118-f001:**
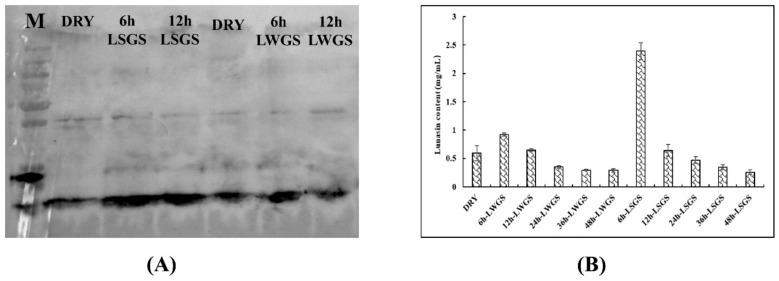
(**A**) Western blot analysis of lunasin expression; (**B**) enzyme-linked immunosorbent assay. Data are displayed as average of three independent experiments, with lines representing ± SD.

**Figure 2 foods-09-00118-f002:**
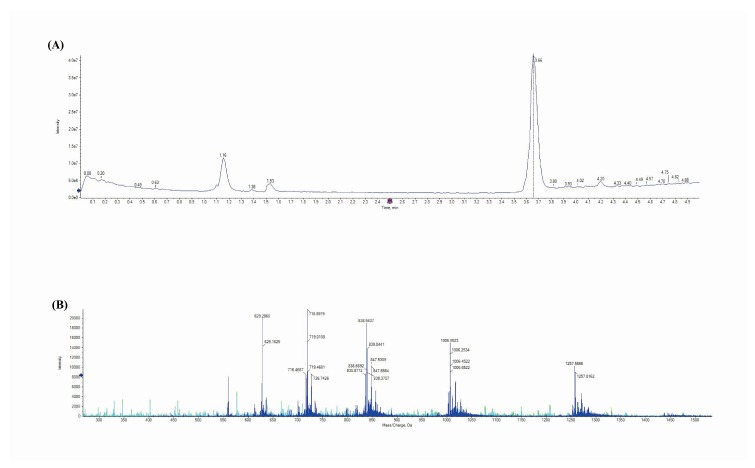
(**A**) UPLC ((Ultra Performance Liquid Chromatography) analysis; (**B**) mass spectrum. The arrow in Figure 2A indicate that the peak area above 2.0e6 will be displayed. The arrow in Figure 2B indicate that ion fragments with a strength of more than 8000 will be displayed.

**Figure 3 foods-09-00118-f003:**
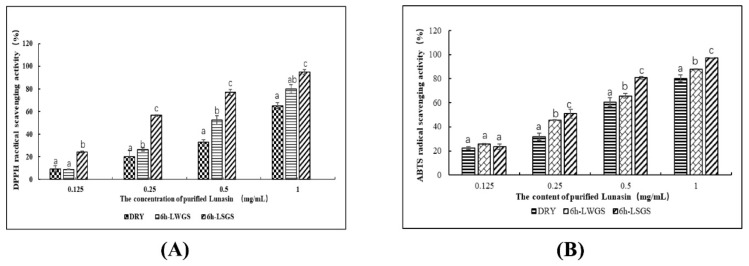
Antioxidant activity assay. (**A**) 1,1-diphenyl-2-picrylhydrazyl radical (DPPH) radical assay; (**B**) 2,2-azino-bis (3-ethylbenzothiazoline-6-sulfonicacid) diammonium salt (ABTS^+^) radical assay. Values are mean ± SD from three experiments. Different letters on bars indicate statistically significant differences (*p* < 0.05).

**Figure 4 foods-09-00118-f004:**
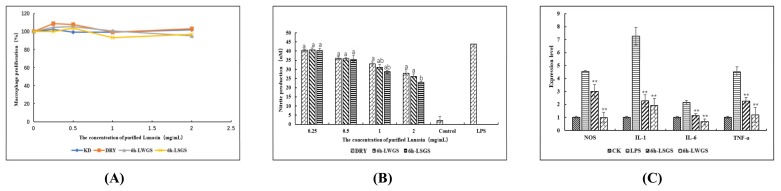
(**A**) Cytotoxicity assay. (**B**) NO release from RAW264.7 cells inhibited by DRY, 6 h-LWGS, and 6 h-LSGS. (**C**) *NOS*, IL-1, IL-6, and TNF-α gene expression in RAW264.7 cells inhibited by DRY, 6 h-LWGS, and 6 h-LSGS. Values are mean ± SD from three experiments; * *p* < 0.05 and ** *p* < 0.01 indicate that there were significant differences in DRY, 6 h-LWGS, and 6 h-LSGS. Different letters on bars indicate statistically significant differences (*p* < 0.05).

**Figure 5 foods-09-00118-f005:**
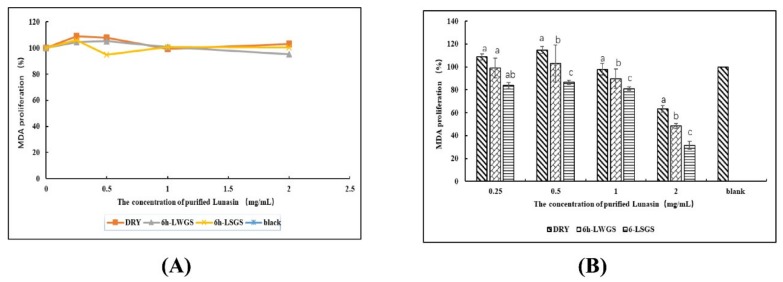
(**A**) Cytotoxicity assay. (**B**) Proliferation of MDA-MB-231 cells inhibited by DRY, 6 h-LWGS, and 6 h-LSGS. Values are mean ± SD from three experiments. Different letters above the columns indicate statistically significant differences (*p* < 0.05). Different letters on bars indicate statistically significant differences (*p* < 0.05).

**Table 1 foods-09-00118-t001:** Primers and sequences.

Primers	Sequences
Actin-F	CCATCATGAAGTGTGACGTTG
Actin-R	ATCTCCTTCTGCATCCTGTCA
IL1b-F	ACTGTGAAATGCCACCTTTTG
IL1b-R	TTTGAAGCTGGATGCTCTCAT
IL6-F	TCAATTCCAGAAACCGCTATG
IL6-R	TTGGGAGTGGTATCCTCTGTG
Nos2-F	GTCCGAAGCAAACATCACATT
Nos2-R	TGAGGGCTCTGTTGAGGTCTA
Tnf-F	GGTTCTCTTCAAGGGACAAGG
Tnf-R	GGCAGAGAGGAGGTTGACTTT

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
