# Peer review of "Effects of Salt Stimulation on Lunasin Accumulation and Activity during Soybean Germination"

_foods, 2020, doi:10.3390/foods9020118_

Round 1

Reviewer 1 Report

General comments - to read carefully and respond, where applicable

The article deals with the sensitivity of the lunasin content and activity in soybean to germination under salt stress conditions. The activity was investigated with regard to the antioxidant capacity, the anti-inflammatory activity, and the anti-carcinogenic (anti-metastatic) function.

Although lunasin, as a bioactive peptide, has been extensively studied (and patented), either from soybean or other biological resources, the effect of salt stress appears sufficiently original.

The study was fairly well conceived, and the takeaways of the study are interesting and relevant. However, the study is quite limited in the sensitivity analysis: an only concentration level of salt (that I could not find, by the way), and only few germination times (0 -6 -12 - 24 - 36 - 48 hours), with the significant results at 6 hours - indeed, with a striking peak in the level of lunasin content at 6 hours, which raises questions about the optimization of the process: maybe, that level would be even greater at 3 hours, or 9 hours, just for example.

At least, in Section 2, the Authors should explain the reason for the above-mentioned limitations to their analysis, and - possibly in Section 4 - recommend directions for further research, for example using different salt concentrations, as well as more frequent germination times around the 6-hours peak.

The language is fair enough, yet several words and statements are wrong and must be fixed.

Specific comments

Line 14. "evaluate the effects of salt stimulation". Few words to explain why trying salt stimulation, such as "based on previous studies", or "for the first time" - clarify the motivation.

Line 28. "to enhance lunasin accumulation in soybean". Change to "to enhance lunasin accumulation and its properties in soybean".

Line 39. "decomposing". Change to "decomposition".

Line 40. "reachs". Change to "reaches".

Line 46. "light treatment, temperature". Change to "light and temperature treatments".

Lines 61-62. "This study not only evaluates the effects of salt stress on lunasin content and activity in the soybean germination process". Here the Authors should claim the novelty, if applicable: was this the first time that salt stress was applied to soybean germination, aimed at increasing lunasin content and activity?

Line 73. "2.2 Soybean Germination". Maybe I have missed the point, anyway: what is the concentration of salt in salt-stress test?

Line 77. "into a Petri". Remove "a".

Line 83. "0.1 mol L−1,". Change to "0.1 mol L−1,".

Line 87. "faction". Change to "fraction".

Lines 135-136. "was then mixed with the extraction solution in the samples were kept in darkness for 6 minutes at room temperature". Unclear, to be reworded.

Line 136. "absorbance". Repeated word.

Line 148. "Nitric oxide (no)". Change to "Nitric oxide (NO)".

Lines 180-181. "This preliminary suggest". Change to "This suggests".

Line 188. "increased or decreased". Change to "increase or decrease".

Line 189. "altered in regulation". Change to "altered regulation".

Line 190."It may also because of inhibition". Change to "It may also be due to the inhibition".

Line 192. "Park et al...". Reference needed.

Line 199. "3.2 Mass Spectrometry Analysis". Explain the added value brought in by the mass spectrometry analysis.

Line 200. "In this study,". Remove.

Line 203. "previously". Change to "previous".

Line 208. "In this study,". Remove.

Line 215. "Comprehensively, salt treatment resulted to". Change to "Overall, salt treatment led to".

Line 223. "as against control mainly". Change to: "in comparison to the control might be".

Line 235. "monitor". Change to "monitoring".

Line 239. "manne". Change to "way".

Line 244. "than that of". Change to "in comparison to the".

Line 245. "result from". Change to "the result of".

Line 246. "inhibite the gene expressions". Change to "as well as the gene expressions".

Line 252. "and found". Change to "as well as".

Line 284."4. Conclusions". This section is far too short and generic, and must be completely reworked. The authors should convey the main takeways from this study, which, by the way, are important and relevant, although deriving from a quite limited experimental design. This is also a good point for recommendations for further research (see general comments).

Lines 285-286. "This section is not mandatory, but can be added to the manuscript if the discussion is unusually long or complex.". Remove.

Line 287. "ambibition". Change to "imbibition".

Author Response

Thanks for the reviewers’ comments and suggestions concerning our manuscript. Those comments are the important guiding to our researches, contribute to improving our paper. We have investigated the comments carefully and made the correction.

Response to Reviewer 1 Comments:

The study was fairly well conceived, and the takeaways of the study are interesting and relevant. However, the study is quite limited in the sensitivity analysis: an only concentration level of salt (that I could not find, by the way), and only few germination times (0 -6 -12 - 24 - 36 - 48 hours), with the significant results at 6 hours - indeed, with a striking peak in the level of lunasin content at 6 hours, which raises questions about the optimization of the process: maybe, that level would be even greater at 3 hours, or 9 hours, just for example.

Thanks for suggestions. Actually, we chose germination for 6 h just because of the radicle emergence at 6-hour soybean. We should chose the more frequent germination times around the 6-hours peak. So we re-germinated and sampled 0, 2, 4, 6, 8, 10, 12 h. The content of lunasin in germinated soybean was measured by enzyme-linked immunoassay. The germination assay showed that lunasin content was relative higher when the radicle emerged. We speculate that it may be due to the need for the germination, the related proteins were gradually synthesis. When the radical begins to grow, the proteins was catabolic for the acceleration growth, thus, lunasin content gradually decreased.

Compared with the control, 50 mM NaCl affected soybean germination, but only slightly delayed germination time, and did not have a severe impact on the final germination percentage. However, soybean germination was significantly inhibited by exposure to 100 mM NaCl, and the soybean growing very poorly, which could not meet the need for developing the germinating functional food. Finally, a low concentration of 50 mM was selected to stimulate the soybeans.

Point 1: Line 14. "evaluate the effects of salt stimulation". Few words to explain why trying salt stimulation, such as "based on previous studies", or "for the first time" - clarify the motivation.

Response 1: Based on previous studies,salt stress can induce changes in many polypeptides and translatable mRNA levels in plants.

Point 2: Line 28. "to enhance lunasin accumulation in soybean". Change to "to enhance lunasin accumulation and its properties in soybean".

Response 2: Thanks, it has been revised.

Point 3: Line 39. "decomposing". Change to "decomposition".

Response 3: Thanks, it has been revised.

Point 4: Line 40. "reachs". Change to "reaches".

Response 4: Thanks, it has been revised.

Point 5: Line 46. "light treatment, temperature". Change to "light and temperature treatments".

Response 5: Thanks, it has been revised.

Point 6: Lines 61-62. "This study not only evaluates the effects of salt stress on lunasin content and activity in the soybean germination process". Here the Authors should claim the novelty, if applicable: was this the first time that salt stress was applied to soybean germination, aimed at increasing lunasin content and activity?

Response 6: This study not only evaluates the effects of salt stress on lunasin content and activity in the soybean germination process but also identifies the key germination time point of more lunasin accumulation by salt stimulation. It was the first time that salt stress was applied to soybean germination, aimed at increasing lunasin content.

Point 7: Line 73. "2.2 Soybean Germination". Maybe I have missed the point, anyway: what is the concentration of salt in salt-stress test?

Response 7: Washed high-quality soybean seeds that were the same size and without decolorization were soaked in 25 °C water or 50mmol/L NaCl solution for 1 hours and then transferred into a Petri dishes placed in a thermostat dark house.

Point 8: Line 77. "into a Petri". Remove "a".

Response 8: Thanks, it has been deleted.

Point 9: Line 83. "0.1 mol L−1,". Change to "0.1 mol L−1,".

Response 9: Thanks, it has been changed.

Point 10: Line 87. "faction". Change to "fraction".

Response 10: Thanks, it has been changed.

Point 11: Lines 135-136. "was then mixed with the extraction solution in the samples were kept in darkness for 6 minutes at room temperature". Unclear, to be reworded.

Response 11: ABTS+ solution was then mixed with the extraction solution in the samples and were kept in darkness for 6 minutes at room temperature.

Point 12: Line 136. "absorbance". Repeated word.

Response 12: Thanks, it has been deleted.

Point 13: Line 148. "Nitric oxide (no)". Change to "Nitric oxide (NO)".

Response 13: Thanks, it has been changed.

Point 14: Lines 180-181. "This preliminary suggest". Change to "This suggests".

Response 14: Thanks, it has been changed.

Point 15: Line 188. "increased or decreased". Change to "increase or decrease".

Response 15: Thanks, it has been changed.

Point 16: Line 189. "altered in regulation". Change to "altered regulation".

Response 16: Thanks, it has been changed.

Point 17: Line 190."It may also because of inhibition". Change to "It may also be due to the inhibition".

Response 17: Thanks, it has been changed.

Point 18: Line 192. "Park et al...". Reference needed.

Response 18: Thanks, it has been changed.

Point 19: Line 199. "3.2 Mass Spectrometry Analysis". Explain the added value brought in by the mass spectrometry analysis.

Response 19: To further confirm the results of ELISA and western, lunasin was indeed existed in the samples.

Point 20: Line 200. "In this study,". Remove.

Response 20: Thanks, it has been changed.

Point 21: Line 203. "previously". Change to "previous".

Response 21: Thanks, it has been changed.

Point 22: Line 208. "In this study,". Remove.

Response 22: Thanks, it has been changed.

Point 23: Line 215. "Comprehensively, salt treatment resulted to". Change to "Overall, salt treatment led to".

Response 23: Thanks, it has been changed.

Point 24: Line 223. "as against control mainly". Change to: "in comparison to the control might be".

Response 24: Thanks, it has been changed.

Point 25: Line 235. "monitor". Change to "monitoring".

Response 25: Thanks, it has been changed.

Point 26: Line 239. "manne". Change to "way".

Response 26: Thanks, it has been changed.

Point 27: Line 244. "than that of". Change to "in comparison to the".

Response 27: Thanks, it has been changed.

Point 28: Line 245. "result from". Change to "the result of".

Response 28: Thanks, it has been changed.

Point 29: Line 246. "inhibite the gene expressions". Change to "as well as the gene expressions".

Response 29: Thanks, it has been changed.

Point 30: Line 252. "and found". Change to "as well as".

Response 30: Thanks, it has been changed.

Point 31: Line 284."4. Conclusions". This section is far too short and generic, and must be completely reworked. The authors should convey the main takeways from this study, which, by the way, are important and relevant, although deriving from a quite limited experimental design. This is also a good point for recommendations for further research (see general comments).

Response 31: This paper evaluated that the effects of salt stimulation on lunasin accumulation and activity during soybean germination, 50 mM NaCl was applied to soybean germination with water treatment as the control group. Lunasin content increased gradually in control group during the germination, and reached the highest level after 6 h imbibition and then decreased slowly. Under salt exposure, lunasin content also exhibited a trend of increasing and then decreasing. It is worth noting that lunasin content in germinating soybean under salt stimulation is significantly higher than that of control group. Salt-stimulation (6h-LSGS) was 2.5 times of the control group(6h-LWGS). Moreover, 6h-LSGS exhibited stronger antioxidant, anti-inflammatory and anti-cancer activity than 6h-LWGS. Overall, this study offered a feasible elicitation strategy to enhance lunasin accumulation in soybean for possible use in the functional food.

Point 32: Lines 285-286. "This section is not mandatory, but can be added to the manuscript if the discussion is unusually long or complex.". Remove.

Response 32: Thanks, it has been deleted.

Point 33: Line 287. "ambibition". Change to "imbibition".

Response 33: Thanks, it has been changed.

Reviewer 2 Report

The study presents effects of salt stimulation on lunasin accumulation and activity during soybean germination. After a careful survey, I came to the conclusion that some points need to be further explained or revised.

Specific comments:

The amino acid sequence of lunasin is is redundant in the Introduction. What was the purity of lunasin standard used in experiment?

The peak of the main analyzed compound should be signed in the chromatogram of the (Figure 2A).

Poor quality figures are difficult to read and Did the authors optimize UPLC conditions (temperature, flow rate, gradient programme, injection volume)?

The manuscript doesn't contain validation of chromatographic procedures used in experiment (LOD, LOQ, precision…) Did the authors measure absorbance only after 30 min for DPPH and after 6 min for ABTS+? This approach doesn't allow to monitor changes of absorbance over the time and when plateau will be reached.

Author Response

Thanks for the reviewers’ comments and suggestions concerning our manuscript. Those comments are the important guiding to our researches, contribute to improving our paper. We have investigated the comments carefully and made the correction.

Response to Reviewer 2 Comments

Point 1: The amino acid sequence of lunasin is is redundant in the Introduction. What was the purity of lunasin standard used in experiment?

Response 1: Thanks,the amino acid sequence has been deleted. The purity of lunasin standard is 95%.

Point 2: The peak of the main analyzed compound should be signed in the chromatogram of the (Figure 2A).

Response 2: Sorry, I will upload the clearest picture in the system.The lunasin chromatograms clearly exhibited a peak at the retention time of 3.66 min. Mass spectrum acquired from the peak at 3.66 min generated the [M + 7H]7+ at 718.90 m/z, [M + 6H]6+ at 838.54 m/z, [M + 5H]5+ at 1006.45, and [M + 4H]4+ at 1257.39 m/z.

Point 3: Poor quality figures are difficult to read and Did the authors optimize UPLC conditions (temperature, flow rate, gradient programme, injection volume)?

Response 3: Sorry, I will upload the clearest picture in the system. Chromatographic analysis was performed at 30 °C using an ACQUITY UPLC ®BEH Shield RP18 (1.7 μm, 2.1 mm×100 mm, Waters Corp., Milford, MA, USA) and a gradient system with the mobile phase composed of buffer A (0.1% formic acid in water) and buffer B (0.1% formic acid in acetonitrile). The flow rate was 0.2 mL·min−1.The gradient programme used was: 1 min 95% A, 2 min 65% A, 4 min 20% A, and 5 min 95% A. The injection volume was 10 µL. MS data were collected from 0 to 5 min.

Point 4: The manuscript doesn't contain validation of chromatographic procedures used in experiment (LOD, LOQ, precision…)

Response 4: Signal-to-Noise-Ratio, SNR: 4276.   Standard concentration: 100ug/mL LOD: 70ng/mL. The mass spectrometry analysis was performed for further confirm the results of ELISA and western, lunasin was indeed existed in the samples.

Point 5: Did the authors measure absorbance only after 30 min for DPPH and after 6 min for ABTS+? This approach doesn't allow to monitor changes of absorbance over the time and when plateau will be reached.

Response 5: Thank you very much for the reviewer's careful pointing out my limitation. Before this, we really didn't pay attention to the methods. We have tested the antioxidant activity of lunasin, and found that changes of absorbance over the time. Absorbance of lunaisn reached plateau after 1hour for DPPH and after 2 hour for ABTS+. The methods and results in the manuscript have been modified. Thank you again for your suggestions.

Round 2

Reviewer 1 Report

The Authors responded comprehensively and made all the requested changes to the manuscript. All the responses are convincing, although they admitted that further experiments would have been useful.

Author Response

Point 1: The Authors responded comprehensively and made all the requested changes to the manuscript. All the responses are convincing, although they admitted that further experiments would have been useful. 

Response 1: Thank you for your suggestion. We will conduct a more in-depth and comprehensive study in the future.

Reviewer 2 Report

Authors improved their manuscript according to reviewer's comments. In my opinion in the current form this article is ready for publication.

Author Response

Point 1: Authors improved their manuscript according to reviewer's comments. In my opinion in the current form this article is ready for publication.

Response 1: Thank you for your approval and support.